# NormLime: A New Feature Importance Metric for Explaining Deep Neural Networks

## Abstract

The problem of explaining deep learning models, and model predictions generally, has attracted intensive interest recently. Many successful approaches forgo global approximations in order to provide more faithful local interpretations of the model's behavior. LIME develops multiple interpretable models, each approximating a large neural network on a small region of the data manifold, and SP-LIME aggregates the local models to form a global interpretation. Extending this line of research, we propose a simple yet effective method, NormLIME, for aggregating local models into global and class-specific interpretations. A human user study strongly favored class-specific interpretations created by NormLIME to other feature importance metrics. Numerical experiments employing Keep And Retrain (KAR) based feature ablation across various baselines (Random, Gradient-based, LIME, SHAP) confirms NormLIME's effectiveness for recognizing important features.

## 1 Introduction

As the applications of deep neural networks continue to expand, the intrinsic black-box nature of neural networks creates a potential trust issue. For application domains with high cost of prediction error, such as healthcare Phan et al. (2017), it is necessary that human users can verify that a model learns reasonable representation of data and the rationale for its decisions are justifiable according to societal norms (Koh & Liang, 2017; Fong & Vedaldi, 2018; Zhou et al., 2018; Lipton, 2016; Langley, 2019).

An interpretable model, such as a linear sparse regression, lends itself readily to model explanation. Yet due to limited capacity, these interpretable models cannot approximate the behavior of neural networks globally. A natural solution, as utilized by LIME Ribeiro et al. (2016), is to develop multiple interpretable models, each approximating the large neural network locally on a small region of the data manifold. Global explanations can be obtained by extracting common explanations from multiple local approximations. However, how to best combine local approximations remains an open problem.

Extending this line of research, we propose a novel and simple feature scoring metric, *NormLIME*, which estimates the importance of features based on local model explanations. In this paper, we empirically verify the new metric using two complementary tests. First, we examine if the Norm-LIME explanations agree with human intuition. In a user study, participants favored the proposed approach over three baselines (LIME, SmoothGrad and VarGrad) with NormLIME receiving 30% more votes than all the baselines combined. Second, we numerically examine if explanations created by NormLIME accurately capture characteristics of the machine learning problem at hand, using the same intuition proposed by Hooker et al. (2018). Empirical results indicate that NormLIME identifies features vital to the classification performance more accurately than several existing methods. In summary, we find strong empirical support for our claim that NormLIME provides accurate and human-understandable explanations for deep neural networks.

The paper makes the following contributions:

- We propose a simple yet effective extension of LIME, called NormLIME, for aggregating interpretations around local regions on the data manifold to create global and class-specific

Figure 1: Explanations for the MNIST digits 9 and 3, generated by: VarGrad (Adebayo et al., 2018), SmoothGrad (Smilkov et al., 2017), LIME (Ribeiro et al., 2016), and NormLIME (ours).

interpretations. NormLIME outperforms LIME and other baselines in two complementary evaluations.

- We show how feature importance from LIME can be aggregated to create class-specific interpretations, which stands between the fine-grained interpretation at the level of data points and the global interpretation at the level of entire datasets, enabling a hierarchical understanding of machine learning models. The user study indicates that NormLIME excels at this level of interpretation.

## 2 RELATED WORK

A machine learning model can be interpreted from the perspective of how much each input feature contributes to a given prediction. In computer vision, this type of interpretation is often referred to as *saliency maps* or *attribution maps*. A number of interpretation techniques, such as SmoothGrad (Smilkov et al., 2017), VarGrad (Adebayo et al., 2018), Integrated Gradients (Sundararajan et al., 2017), Guided Backpropagation (Springenberg et al., 2015), Guided GradCAM (Selvaraju et al., 2017), and Deep Taylor Expansion (Montavon et al., 2017), exploit gradient information, as it provides a first-order approximation of the input's influence on the output (Simonyan et al., 2013; Ancona et al., 2018). Seo et al. (2018) analyze the theoretical properties of SmoothGrad and Var-Grad. When the gradient is not easy to compute, Baehrens et al. (2010) places Parzen windows around data points to approximate a Bayes classifier, from which gradients can be derived. DeepLift (Shrikumar et al., 2017) provides a gradient-free method for saliency maps. Though gradient-based techniques can interpret individual decisions, aggregating individual interpretations for a global understanding of the model remains a challenge.

Local interpretations are beneficial when the user is interested in understanding a particular model decision. They become less useful when the user wants a high-level overview of the model's behavior. This necessitates the creation of global interpretations. LIME (Ribeiro et al., 2016) first builds multiple sparse linear models that approximate a complex model around small regions on the data manifold. The weights of the linear models can then be aggregated, and compared, in order to construct a global explanation using Submodular Pick LIME (SP-LIME). Ribeiro et al. (2018) introduce anchor rules to capture interaction between features. Tan et al. (2018) approximate the complex model using a sum of interpretable functions, such as trees or splines, which capture the influence of individual features and their high-order interactions. The proposed NormLIME technique fits into this "neighborhood-based" paradigm. We directly modify the normalization for aggregated weights, rather than the function forms (such as rules or splines).

Ibrahim et al. (2019) ranks feature importance and clusters data points based on their ranking correlation. The interpretations for cluster medoids are used in the place of a single global interpretation. Instead of identifying clusters, in this paper, we generate interpretations for each class in a given dataset. The class-level interpretation provides an intermediate representation so that users can grasp behaviors of machine learning models at different levels of granularity.

Proper evaluation of saliency maps can be challenging. Adebayo et al. (2018) show that, although some techniques produce visually reasonable saliency maps, such maps may not faithfully reflect the behavior of the underlying model. Thus, visual inspection by itself is not a reliable evaluation criterion. Kindermans et al. (2017) adopt linear classifiers as a sanity check. Feng & Boyd-Graber (2019) propose cooperative games, where humans can see the interpretation of their AI teammate, as a benchmark. Hooker et al. (2018) propose ablative benchmarks for the evaluation of feature importance maps. When features that are considered important are removed from the input, the

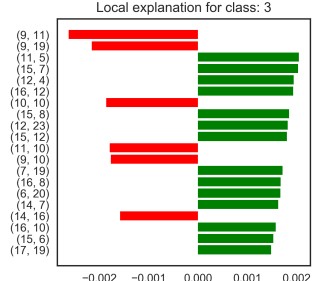 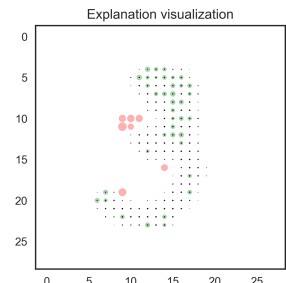

Figure 2: A LIME-based Local Explanation using the input pixels as features. On the right, green and red indicate pixels whose presence/absence offers support for the class label "3". On the left, we show the 20 pixels with the largest weights.

model should experience large drops in performance. The opposite should happen when features deemed unimportant are removed. In this paper, we evaluate the proposed technique using the Keep-and-Retrain (KAR) criterion from Hooker et al..

## 3 METHODOLOGY

As a method for explaining deep models, LIME (Ribeiro et al., 2016) first builds interpretable models where each approximates the complex model around a locality on the data manifold. After that, the local explanations are aggregated using SP-LIME. In this work, we extend the general paradigm of LIME with a new method, which we call NormLIME, for aggregating the local explanations.

### 3.1 BUILDING LOCAL EXPLANATIONS

LIME constructs local interpretable models to approximate the behavior of a large, complex model within a locality on the data distribution. This process can be analogized with understanding how a hypersurface $f(\boldsymbol{x})$ changes around $\boldsymbol{x}_0$ by examining the tangent hyperplane $\nabla f(\boldsymbol{x}_0)$.

Formally, for a given model $f : \mathcal{X} \mapsto \mathcal{Y}$, we may learn an interpretable model $g$, which is local to the region around a particular input $\boldsymbol{x}_0 \in \mathcal{X}$. To do this, we first sample from our dataset according to a Gaussian probability distribution $\pi_{\boldsymbol{x}_0}$ centered around $\boldsymbol{x}_0$. Repeatedly drawing $\boldsymbol{x}'$ from $\pi_{\boldsymbol{x}_0}$ and applying $f(\cdot)$ yield a new dataset $\mathcal{X}' = \{(\boldsymbol{x}', f(\boldsymbol{x}'))\}$. We then learn a sparse linear regression $g(\boldsymbol{x}', \boldsymbol{x}_0) = \boldsymbol{w}_{\boldsymbol{x}_0}^\top \boldsymbol{x}'$ using the local dataset $\mathcal{X}'$ by optimizing the following loss function with $\Omega(\cdot)$ as a measure of complexity.

$$\operatorname*{argmin}_{\boldsymbol{w}_{\boldsymbol{x}_0}} \mathcal{L}(f, g, \pi_{\boldsymbol{x}_0}) + \Omega(\boldsymbol{w}_{\boldsymbol{x}_0}) \tag{1}$$

where $\mathcal{L}(f, g, \pi_{\boldsymbol{x}_0})$ is the squared loss weighted by $\pi_{\boldsymbol{x}_0}$

$$\mathcal{L}(f, g, \pi_{\boldsymbol{x}_0}) = \mathop{\mathbb{E}}_{x' \sim \pi_{\boldsymbol{x}_0}} \left( f(\boldsymbol{x}') - g(\boldsymbol{x}', \boldsymbol{x}_0) \right)^2 \tag{2}$$

For $\Omega(\boldsymbol{w}_{\boldsymbol{x}_0})$, we impose an upper limit $K$ for the number of non-zero components in $w_{\boldsymbol{x}_0}$, so that $\Omega(\boldsymbol{w}_{\boldsymbol{x}_0}) = \infty \cdot 1 \left( \|\boldsymbol{w}_{\boldsymbol{x}_0}\|_0 > K \right)$. The optimization is intractable, but we approximate it by first selecting $K$ features with LASSO regression and performing regression on only the top $K$ features.

This procedure yields $g(\boldsymbol{x}', \boldsymbol{x}_0)$, which approximates the complex model $f(\boldsymbol{x})$ around $\boldsymbol{x}_0$. The components of the weight vector $\boldsymbol{w}_{\boldsymbol{x}_0}$ indicate the relative influence of the individual features of $\boldsymbol{x}$ in the sample $\mathcal{X}'$ and serve as the local explanation of $f(\boldsymbol{x})$. Figure 2 illustrates such a local explanation.

### 3.2 NORMLIME

After a number of local explanations have been constructed, we aim to create a global explanation. NormLIME is a method for aggregating and normalizing multiple local explanations and estimating

the global relative importance of all features utilized by the model. NormLIME gives a more holistic explanation of a model than the local approximations of LIME.

We let $c_i$ denote the $i^{\text{th}}$ feature, or the $i^{\text{th}}$ component in the feature vector $\boldsymbol{x}$. Since the local explanation weights are sparse, not all local explanations utilize $c_i$. We denote the set of local explanations that do utilize $c_i$ as $E(c_i)$, which is a set of weight vectors $\boldsymbol{w}_{x_j}$ computed at different locales $\boldsymbol{x}_j$. In other words, for all $\boldsymbol{w} \in E(c_i)$, the corresponding weight component $w_i \neq 0$.

The NormLIME "importance score" of the feature $c_i$, denoted by $\mathcal{S}(c_i)$, is defined as the weighted average of the absolute values of the corresponding weight $\boldsymbol{w}_i, \forall \boldsymbol{w} \in E(c_i)$.

$$\mathcal{S}(c_i) := \frac{1}{|E(c_i)|} \sum_{\boldsymbol{w}_{x_j} \in E(c_i)} \gamma(\boldsymbol{w}_{x_j}, i) \left| w_{x_j, i} \right|, \tag{3}$$

where the weights $\gamma$ are computed as follows.

$$\gamma(\boldsymbol{w}_{x_j}, i) := \frac{\left| w_{x_j, i} \right|}{\sum_k \left| w_{x_j, k} \right|} = \frac{\left| w_{x_j, i} \right|}{\| \boldsymbol{w}_{x_j} \|_1}. \tag{4}$$

Here, $\gamma(\boldsymbol{w}_{x_j}, i)$ represents the relative importance of the feature $c_i$ in the local model built around the data point $x_j$. If a feature $c_i$ is not utilized in any local models, we set its importance $\mathcal{S}(c_i)$ to 0.

We now introduce a slightly different perspective of NormLIME, which helps us understand the difference between this approach and the aggregation and feature importance approach used in LIME. Consider the global feature weight matrix $M$, whose rows are the local explanation $w$ computed at different locales. Let $\boldsymbol{\omega}_i$ be the $i^{\text{th}}$ column of the matrix, which contains the weights for the same feature in different local explanations. Let $\boldsymbol{v}$ be the vector representing the L1 norms of the rows of M. We can express the NormLIME global feature importance function as

$$\mathcal{S}(c_i) = \frac{1}{\| \boldsymbol{\omega}_i \|_0} \boldsymbol{\omega}_i^\top \text{diag}(\boldsymbol{v}^{-1}) \boldsymbol{\omega}_i, \tag{5}$$

where $\text{diag}(\boldsymbol{v}^{-1})$ denotes a matrix with $\boldsymbol{v}^{-1}$ on the diagonal and zero everywhere else. $\| \cdot \|_0$ is the L0 norm, or the number of non-zero elements.

In comparison, the submodular pick method (SP-LIME) by Ribeiro et al. (2016) employs the L2 norm

$$I^{SP}(c_i) = \| \boldsymbol{\omega}_i \|_2 \tag{6}$$

to measure the importance of $c_i$. Contrasting Eq. equation 5 and equation 6, it is apparent that the difference between the two methods lies in in the normalization of the column weights.

## 3.3 CLASS-SPECIFIC INTERPRETATIONS

As discussed above, NormLIME estimates the overall relative importance assigned to feature $c_i$ by the model. For binary classification problems, this is equivalent to a representation of the importance the model assigns to the feature $c_i$ in distinguishing between the two classes, i.e., recognizing a class label. In multi-class problems, however, this semantic meaning is lost as the salience computation above does not distinguish between classes, and class-relevant information becomes muddled together.

It is straightforward to recover the salience information associated with individual class labels, by partitioning $E(c_i)$ based on the class label of the initial point $x_j$ that the local approximation is built around. The partition $E_y(c_i)$ contains the local explanation $w_{x_j}$ if and only if $f(\cdot)$ assigns the label $y$ to $x_j$. More formally,

$$E_y(c_i) = \{ w_{x_j} \in E(c_i) \mid f(x_j) = y \}. \tag{7}$$

It is easy to see that if $E_y(c_i) \neq \varnothing, \forall y$, and $f(x)$ is a single-label classification problem, then the family of sets $E_y(c_i)$ forms a partition of $E(c_i)$:

$$E(c_i) = \bigcup_y E_y(c_i), \quad E_{y_j}(c_i) \cap E_{y_k}(c_i) = \varnothing \; \forall y_j \neq y_k. \tag{8}$$

Computing salience of $c_i$ for a given label is performed via

$$\mathcal{S}_y(c_i) := \frac{1}{|E_y(c_i)|} \sum_{\boldsymbol{w}_{\boldsymbol{x}_j} \in E_y(c_i)} \gamma(\boldsymbol{w}_{\boldsymbol{x}_j}, i) \left| w_{x_j, i} \right|. \tag{9}$$

Compared to global interpretations, the class-specific salience $\mathcal{S}_y(c_i)$ yields higher resolution information about how the complex model differentiates between classes. We use $\mathcal{S}_y(c_i)$ as prediction-independent class-level explanations in the human evaluation, described in the next section.

## 4 HUMAN EVALUATION

In order to put the interpretations generated by NormLIME to test, we administered a human user study across Amazon Mechanical Turk users. We compare class-specific explanations generated by standard feature importance functions and the proposed salience method on the MNIST dataset.

Consider Figure 3:

*Top:* an example question from the user study.
*Bottom:* an example attention check from the user study. The correct choice is option 2. Options 1 and 4 are duplicates and cannot distinguish well between labels "3" and "8".

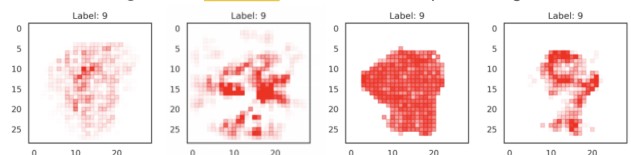

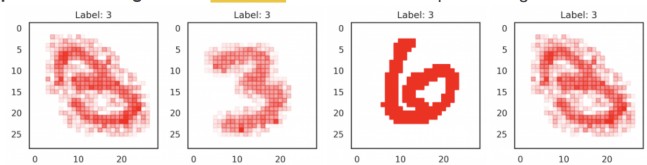

Figure 3: Example survey questions.

### 4.1 SALIENCY MAP BASELINES

To avoid showing too many options to the human participants, which may cause cognitive overload, we selected a few baseline techniques that we consider to be the most promising for saliency maps (Seo et al., 2018). We select SmoothGrad and Var-Grad because they aim to reduce noise in the gradient, which should facilitate the aggregation of individual decisions to form a class-level interpretation. The aggregation of these individual interpretations are performed by taking the mean importance scores over interpretations from a sample of datapoints corresponding to each label. The details are discussed below.

#### 4.1.1 SMOOTHGRAD

This technique (Smilkov et al., 2017) generates a more "interpretable" salience map by averaging out local noise typically present in gradient interpretations. We add random noise $\eta \sim \mathcal{N}(0, \sigma^2)$ to the input $x$. Here we follow the default implementation using the "SmoothGrad Squared" formula, which is an expectation over $\eta$:

$$I^{SGSQ}(x) = \mathbb{E}\left[ |\frac{\partial f(x + \eta)}{\partial c_k}|^2 \right]$$

as noted in Hooker et al. (2018). In practice, we approximate the expectation with the average of 100 samples of $\eta$ drawn from $\mathcal{N}(0, \sigma^2)$ where $\sigma$ is 0.3. The class-level interpretation is computed as the average of the saliency maps for 10 images randomly sampled from the target class.

### 4.1.2 VARGRAD

Similar to SmoothGrad, VarGrad (Adebayo et al., 2018) uses local sampling of random noise $\eta \sim \mathcal{N}(0, \sigma^2)$ to reduce the noise of the standard gradient map interpretation. VarGrad perturbs an input $x$ randomly via $\eta$, and then computes the component-wise variance of the gradient over the sample

$$I^{VG}(x) = \text{Var}(|\frac{\partial f(x + \eta)}{\partial c_k}|).$$

Similar to SmoothGrad, we compute the variance using 100 samples of $\eta$ from a normal distribution with zero mean and standard deviation of 0.3. We use the average saliency map over 10 randomly sample images in the desired class as the class-level interpretation.

### 4.1.3 LIME

We compute importance as $I^{SP}(x)$ as in Eq. equation 6, but conditioned on the label $y$ to capture feature that were positively correlated with the specific label: $I^{SP}(x \cdot \mathbf{1}_{x>0})$.

For both LIME and NormLIME, we only show the input features that are positively correlated with the prediction. That is, when $I^{SP}(c_i)$ or $\mathcal{S}(c_i)$ is positive. The purpose is to simplify the instructions given to human participants and avoid confusion, since most participants may not have sufficient background in machine learning.

## 4.2 EXPERIMENTAL DESIGN

The design of the study is as follows: We administered a questionnaire featuring 25 questions, each containing four label-specific explanations for the same digit. We were able to restrict participants through Mechanical Turk to users who had verified graduate degrees. Survey takers were instructed to evaluate the different explanations based on how well they captured the important characteristics of each digit in the model's representation. To account for response ordering bias, the order of the methods presented for each question was randomized. In order to catch participants who cheat by making random choices, we included 5 attention checks with a single acceptable answer that is relatively obvious. We only include responses that pass at least 4 of the 5 attention checks.

We conducted the experiment on MNIST with $28 \times 28$ single-channel images. We trained a 5-layer convolutional network that achieves 99.05% test accuracy. This model consisted of two blocks of convolution plus max-pooling operations, followed by three fully connected layers with dropout in-between, and a final softmax operation. The number of hidden units for the three layers was 128, 128, and 10, respectively. Class-specific explanations were generated for the digits from 0 to 9.

It is important to note that none of the explanations generated for the study represented a particular prediction on a particular image, but instead represented how well the importance functions captured the important features for a label (digit) in the dataset.

## 4.3 RESULTS AND DISCUSSION

After filtering responses that failed the attention check, we ended up with 83 completed surveys. From their responses, the number of votes for each method were: 939 for NormLIME, 438 for LIME, 151 for VarGrad, and 132 for SmoothGrad. We analyzed the data by examining each user's response as a single sample of the relative proportions of the various explanation methods for that user, and performed a standard one-way ANOVA test against the hypothesis that the explanations were preferred uniformly. We obtained a statistically significant result, with an F statistic of 338 and a p-value of $5.22 \times 10^{-100}$, allowing us to reject the null hypothesis. We conclude that a significant difference exist between how users perceived the explanations.

A subsequent Tukey HSD post hoc test confirms that the differences between NormLIME and *all* other methods are highly statistically significant ($p < 0.001$). It also shows that the difference between LIME and the gradient-based interpretations is statistically significant ($p < 0.001$). We conclude that overall, the NormLIME explanations were preferred over all other baseline techniques, including LIME and that NormLIME and LIME were preferred over the gradient-based methods.

Observing Figure 1, the interpretations of SmoothGrad and VarGrad do not appear to resemble anything semantically meaningful. This may be attributed to the fact that these methods are not

designed with class-level interpretation in mind. LIME captures the shape of the digits to some extent but cannot differentiate the most important pixels. In contrast, the normalization factor in NormLIME helps to illuminate the differences among the important pixels, resulting in easy-to-read interpretations. This suggests that proper normalization is important for class-level interpretations.

## 5 NUMERICAL EVALUATION WITH KAR

Visual inspection alone may not be sufficient for evaluating saliency maps (Adebayo et al., 2018). In this section, we further evaluate NormLIME using a technique akin to Keep And Retrain (KAR) proposed by Hooker et al. (2018). The underlying intuition of KAR is that features with low importance are less relevant to the problem under consideration. This gives rise to a principled "hypothesis of least damage": removal of the least important features as ranked by a feature importance method should impact the model's performance minimally. Thus, we can compare two measures of feature importance by comparing the predictive performance after removing the same number of features as ranked by each method as the least important.

Specifically, we first train the same convolutional network as in the human evaluation with all input features and use one of the importance scoring method to rank the features. We remove a number of least important features and retrain the model. Retraining is necessary as we want to measure the importance of the removed features to prediction rather than how much one trained model relies on the removed features. After that, we measure the performance drop caused by feature removal; a smaller performance drop indicates more accurate feature ranking from the interpretation method.

We perform KAR evaluation on two set of features. The first set is the raw pixels from the images. The second set of features are the output from the second convolutional layers of the network. The baselines and results are discussed below.

### 5.1 BASELINES

We evaluated NormLIME against various baseline interpretation techniqes on the MNIST dataset (LeCun, 1998). For NormLIME and LIME, we use the absolute value of $I(c_i)$ as the feature importance. In addition to those used in the human evaluation, we introduce the following baselines.

#### 5.1.1 SHAP

The Shapley value measures the importance of a feature by enumerating all possible combinations of features and compute the average of performance drop when the feature is removed. The value is well suited to situations with heavy interactions between the features. While theoretically appealing, the computation is intractable. A number of techniques (Chen et al., 2019; Lundberg & Lee, 2017; Strumbelj & Kononenko, 2010) have been proposed to approximate the Shapley value. Here we use SHapley Additive exPlanations (SHAP), an approximation based on sampled least-squares (Lundberg & Lee, 2017).

#### 5.1.2 RANDOM

This baseline randomly assigns feature importance. This serves as a "sanity check" baseline. Notable, in the experiments of Hooker et al. (2018), some commonly used saliency maps perform worse than random.

### 5.2 RESULTS AND DISCUSSION

Figure 4 (a) shows the error gained after a number of least important features are removed, averaged over 5 independent runs. We use removal thresholds from 10% to 90%. When 50% of the features or less are removed, NormLIME performs better or similarly with the best baselines, though it picks up more error when more features are removed. The best prediction accuracy among all is achieved by NormLIME at 50% feature reduction with 0% error gain. This is matched by SHAP also at 50% feature reduction and VarGrad at 20% feature reduction. All other baselines observe about at least 0.25% error gain at 50% feature reduction,. SHAP and LIME perform better than other methods,

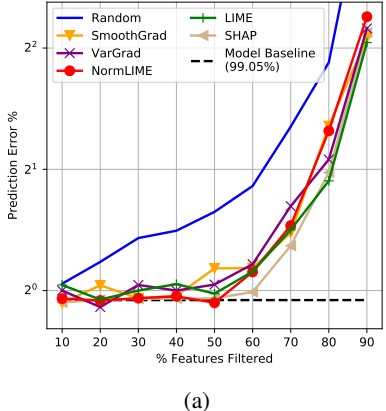 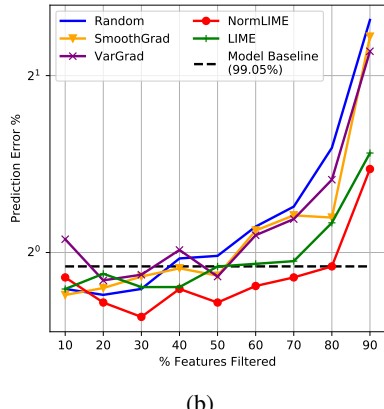

(a)                              (b)

Figure 4: Prediction errors on MNIST as the least important features are gradually removed from: (a) the input. (b) the output of the last convolutional layers. The horizontal axis indicate the percentage of of features removed. The vertical axis shows absolute error in log scale.

including NormLIME, when 60% or more features are removed. The gradient-based methods, are outperformed by NormLIME and LIME.

Figure 4 shows the same measure on the convolutional features. On these features, NormLIME outperforms the other methods by larger margins, compared to the input features. NormLIME achieves better results than the original model (at 99.05%) when 70% or less features are removed, underscoring the effectiveness of dimensionality reduction. The best performance is achieved at 30% removal with a classification accuracy of 99.31%. The second best is achieved by LIME at 99.1% accuracy when filtering 40% of features, comparable with NormLIME performance at the same level.

When 80% of features are removed, NormLIME demonstrates zero error gain, whereas the second best method, LIME, gains 0.3% absolute error. When 90% of features are removed, NormLIME shows 0.45% error gain while LIME observes .6% error gain and all others receive at least 1.25%.

Overall, gradient ensemble methods SmoothGrad and VarGrad outperformed Random but compared unfavorably with "additive local model approximation" schemes (SHAP, LIME, and NormLIME).

The advantage of NormLIME is more pronounced when pruning the convolutional features than input features. Further, we can achieve better performance removing convolutional features, but not the input features. This suggests that there is more redundancy in the convolutional features and NormLIME is able to exploit that phenomenon.

## 6 CONCLUSIONS

Proper interpretation of deep neural networks is crucial for state-of-the-art AI technologies to gain the public's trust. In this paper, we propose a new metric for feature importance, named NormLIME, that helps human users understand a black-box machine learning model.

We extend the LIME / SP-LIME technique (Ribeiro et al., 2016), which generates local explanations of a large neural network and aggregates them to form a global explanation. NormLIME adds proper normalization to the computation of global weights for features. In addition, we propose label-based NormLIME, which provides finer-grained interpretation in a multi-class setting compared to SP-LIME which focuses on selecting an optimal selection of individual predictions to explain a model.

Experimental results demonstrate that the NormLIME explanations agree with human intuition. The human evaluation study shows that explanations generated by NormLIME are strongly favored over comparable ones generated by LIME, SmoothGrad, and VarGrad with strong statistical significance. Further, using Keep-And-Retrain evaluation, we show that explanations formed by the NormLIME metric are faithful to the problem at hand, as it identifies input features and convolutional features whose removal is not only harmless, but may even improve prediction accuracy.

ACKNOWLEDGMENTS

Intentionally omitted to anonymize the manuscript.

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
