# OpenReview forum: "NormLime: A New Feature Importance Metric for Explaining Deep Neural Networks"
_ICLR.cc/2020/Conference — Reject_

### Official Review · AnonReviewer1 · 2019-10-21
**Official Blind Review #1**

**Rating:** 3

**Review:**


This paper proposes a novel scheme to aggregate local explanations
from LIME to generate global and/or class-level interpretations in the
form of feature importances. The technical novelty lies in the
normalization of the feature weights across local explanations where
the proposed scheme utilizes a well-motivated L0 normalization in
place of the existing L2 normalization in Ribeiro et al., 2016. The
sparsity induced by the L0 normalization leads to more "easy-to-read"
global (or class level) "explanations" since the number of features
with non-zero importances is reduced.


The simple idea of generating sparse aggregations of local
explanations to obtain global (or class-level) interpretations is well
motivated and seems fairly intuitive (and I consider simplicity a
strength) and the extremely favourable empirical results, I am leaning
towards a reject for the current version of the paper. The main
reasons for this decision are as follows:

- The proposed scheme is presented as closely tied to LIME, which
  severely limits the scope of the proposed solution. However, I can
  imagine that the high level idea of sparse aggregation of local
  explanations can easily generalize to many local explanation
  schemes.
- The empirical evaluation needs to be improved (or better clarified)
  in my opinion. There are many potential baselines that are also not
  considered (see details below). Moreover, the evaluation is
  completely limited to a single data set (which is good as a strawman
  but not sufficient to make significant claims).


Clarifications re: baselines and empirical evaluations:

- No comparison to Ibrahim, et al., 2019 -- you can cluster in
  unsupervised setting but you can also aggregate within class in
  supervised setting. Generating class-level interpretations is an
  interesting idea but its novelty is somewhat unclear.
- In terms of explanations, there are contrastive explanations [1]
  which are not really discussed in this paper. This scheme could be
  more useful for generating class level interpretations.
- It is not clear why the negative correlations were removed in the
  user study. In certain situations, such as when distinguishing
  between digits 3 & 8, the absence of weights (potentially
  corresponding to negative weights) can be crucial explanations.
- It is not clear why in the SmoothGrad and VarGrad baselines, the
  class level interpretations are generated with just 10 images per
  class. Are the class level interpretations of LIME and NormLIME also
  generated from just 10 images per class. If that is the case, 10
  seems like a very small number and can potentially create noisy
  explanations. If LIME and NormLIME used more than 10 images to
  generate class level interpretations, the comparison does not seem
  fair. Please explain this discrepancy.
- For the evaluation in Sec. 5, it is not clear how the features are
  removed. Are they removed based on the global explanations (feature
  importances) of each baseline? If this is the case, are the global
  explanations of each baseline generated using the same amount of
  local explanations? Please clarify.
- The KAR analysis results in Fig. 4(b) need to be further
  investigated. First, it is not clear if this difference between the
  different baselines something that happens in multiple datasets or
  just this one. Moreover, it seems natural that something like
  SmoothGrad and VarGrad would be able to capture the feature
  redundancies if the redundancies were the actual cause for this
  marked difference between the relative performances in Fig. 4(a) and
  4(b). This could definitely use a better discussion.


[1] Dhurandhar, Amit, et al. "Explanations based on the missing:
Towards contrastive explanations with pertinent negatives." Advances
in Neural Information Processing Systems. 2018.

**Experience Assessment:**

I have published one or two papers in this area.

**Review Assessment: Checking Correctness Of Derivations And Theory:**

N/A

**Review Assessment: Checking Correctness Of Experiments:**

I assessed the sensibility of the experiments.

**Review Assessment: Thoroughness In Paper Reading:**

I read the paper at least twice and used my best judgement in assessing the paper.

---

> ### Author Response · Authors · 2019-11-15
> **Addressing the comments raised by #1**
>
> To address the comments first: Yes it is possible to apply this method to any generic explainer. We haven’t experimented, but it is similar to a weighted version of SmoothGrad except the local models are not localized to the same point. Hence the resulting interpretation is no longer local. Regarding the choice of LIME, LIME has a few specific benefits many other methods lack: (1) LIME explanations are already somewhat less local than a simple gradient, and (2) LIME doesn't require differentiable input features. Of the two, (1) motivated considering LIME interpretations as candidate local models in the more general aggregation approach.
>
> - Regarding the empirical evaluation, Ibrahim et. al.'s approach is not readily adapted for the task of class-level interpretation and we don't want to attribute our own modification, which may or may not work well, to these researchers.
>
> - For the user experiment, it is necessary to provide a limited number of options for the human user to choose from. Too much information will likely cause information overload and increase the likelihood the workers will stop paying attention after a while. Considering the general unfamiliarity with the notion of feature importance on Mechanical Turk, we choose to remove the negative features in order to avoid information overload and make it a more tractable problem for the participants.
>
> We believe our selections (NormLIME and LIME, SmoothGrad, VarGrad) for the user experiments are specifically motivated to capture important comparisons with a minimum number of interpretation techniques: by comparing the effect of the aggregation scheme (LIME vs NormLIME), and comparing against SmoothGrad, a standard gradient method that's simple yet highly effective at generating qualitatively "good" interpretations. The inclusion of VarGrad is motivated by the finding in "Evaluating Feature Importance Estimates" Hooker et. al. 2018 that VarGrad empirically outperforms random importance assignment (whereas many popular interpretability methods fail this benchmark).
>
> - Regarding the novelty of class-level interpretations: Although we are not aware of other works that consider this problem, we don't claim class-level interpretations to be a novel innovation, but merely one approach for recovering some semantic information from the global importance score. As NormLIME does not generate local interpretations, it is more difficult to generate comparison visualizations. We settled on stratifying the dataset by label in order to create a global interpretation problem where we can compare the semantic information captured by different techniques. We motivate label explanations as visualizing the model's approximate representation of each label.
>
> - Contrastive explanations would be appropriate to add to the discussion of similar work in this paper. Thank you for pointing out the relevant literature on contrastive explanations and we will cite it in the next draft.
>
> - Regarding the number of samples of LIME/NormLIME vs SG and VG, this is a mistaken omission. For all explainers in the human user experiments, we generated each class explanation from aggregating n=10 individual interpretations respectively. Even using only 10 local interpretation for each global interpretation we were still able to find that NormLIME aggregation was able to better capture the overall semantic information in the label visualization compared to the other methods based on human evaluation.
>
> - For the evaluation in sec 5, the bottom k% of features are removed (based on the different global explanations), and the resulting drops in performance on retrained models (with the various feature removals) are compared. The global explanations for all baselines (and LIME/NormLIME) are computed with the same number of local explanations.
>
> - Regarding the results indicated in Fig 4(b), it’s not exactly clear what the cause of the difference in results is and is somewhat of an open problem. Our analysis indicates results that seem consistent with previous findings in "Evaluating Feature Importance Estimates", Hooker et. al.. Furthermore, we note that SHAP, LIME, and NormLIME outperform the gradient based methods generally. Our general belief is that this has to do with the fact that the non-gradient methods we examined, including NormLIME, operate less locally than the gradient, and so the methods give better global importance estimates. Specifically regarding the difference between (a) and (b), we have an intuitive hypothesis that the redundancy is more clear or less variable in the input than in the convolutional features. Certainly, it’s not exactly clear and is an open problem. We tried to give some insight for our thoughts on why the discrepancy exists, without erring into conjecture.

---

### Official Review · AnonReviewer2 · 2019-10-22
**Official Blind Review #2**

**Rating:** 6

**Review:**

Months ago, I read this article on arxiv (https://arxiv.org/pdf/1909.04200.pdf). It is an interesting work that tries to propose a simple yet effective interpretable model. I am not familiar with this research direction, and I try to make an educated guess.

Pros:
-- As the author suggested, the method is simple and effective.
-- The authors conducted user studies to demonstrate that the results generated by their proposed method is strongly favored over previous methods.

Cons:
-- Subscripts in equations need improvement to make them consistent. For example, in Equation (7), we have E_{y}, but in Equation (8), we have E_{Y_j} and E_{Y_k}.
-- Section 4, Figure 3, top, it seems obvious to choose the fourth one to distinguish the number 9? I feel this example is too easy and not convincing enough.

**Experience Assessment:**

I do not know much about this area.

**Review Assessment: Checking Correctness Of Derivations And Theory:**

I assessed the sensibility of the derivations and theory.

**Review Assessment: Checking Correctness Of Experiments:**

I assessed the sensibility of the experiments.

**Review Assessment: Thoroughness In Paper Reading:**

I read the paper at least twice and used my best judgement in assessing the paper.

---

> ### Author Response · Authors · 2019-11-15
> **Addressing the comments raised by #2**
>
> Thank you for your comments. Regarding the notational difference between equation 7 and 8, the $y_j$ and $y_k$ merely represent two different labels (we could have called them $y$ and $y'$ perhaps) which are used in the definition of $E_y$ in equation 7. Regarding Section 4, Figure 3, top, that’s an actual question from the user survey. They were generated randomly after hyperparameter tuning over all class labels for each method. Once we generate the image we can’t go back through and tune them post hoc. It seems the issue is then deficiencies in tuning the hyperparameters for each interpretation generator for the labels. For this kind of problem though, as the same hyperparameters generate interpretations across all classes, it is difficult for all images to be tuned jointly and visualize well. Furthermore, some hyperparameters are tuned jointly over all explainers (for instance, we apply the same level of thresholding to all salience maps for a fair comparison).

---

### Official Review · AnonReviewer4 · 2019-11-04
**Official Blind Review #4**

**Rating:** 6

**Review:**

This paper proposes a new method for DNN interpretability based on LIME, itself based on ensembling of large numbers of low-complexity models called "local explanation models". This method allows to better capture the relative importance of each feature, and is also able to recover the class-specific signals that DNNs use to distinguish between a number of classes.

Score: Weak Accept
While this method seems like a good step forward, I am uncertain about its significance, mostly because of the choice of dataset and because it appears to be a minor change to a well known algorithm.

I think some high-level aspects could be better motivated:
- Why is interpreting black-box models a better avenue than building interpretable models?
- Why should we expect this method to work beyond MNIST? Will it work on large images where translational invariance is a vital aspect? Will it work on non-image data?
- Why does this relatively subtle change from LIME create such a dramatic visual difference? Do the authors have a stronger intuition than what is hinted at in the paper?

Other comments:
- The human evaluation is very interesting, and suggests that this method correlates much better with human judgement than previous ones. One key aspect that is missing, but hinted at, is where this difference comes from. It could be from the proposed change or from the class-specific information. (It should be easy to run KAR to test this?)
- It's not always clear how this should be reproduced. For example, the authors specify the number of samples for SmoothGrad and VarGrad but not LIME nor NormLIME.

**Experience Assessment:**

I have read many papers in this area.

**Review Assessment: Checking Correctness Of Derivations And Theory:**

I assessed the sensibility of the derivations and theory.

**Review Assessment: Checking Correctness Of Experiments:**

I assessed the sensibility of the experiments.

**Review Assessment: Thoroughness In Paper Reading:**

I made a quick assessment of this paper.

---

> ### Author Response · Authors · 2019-11-15
> **Addressing the comments raised by #4**
>
> Thank you for raising these interesting and important questions: It makes
> more sense to focus on interpreting black box models because DL models obtain
> a high level of performance, motivating their use (and necessitating their inter-
> pretation). Existing interpretable models such as linear regression or decision
> tree-based models lack the power of current neural networks. While investigat-
> ing how to train deep learning models with embedded interpretable structure is
> an interesting research avenue, it’s a different and more complex problem than
> what we’ve pursued here.
> We carried out a few experiments that give us intuition this technique will
> work beyond MNIST. In order to generalize the concept of a global interpre-
> tation to be more meaningful, we explore generating salience map at internal
> layers. This reduces the impact of requiring translation invariance amongst fea-
> tures. Instead of highlighting the most globally meaningful pixels, it picks out
> the most useful convolutional feature maps. In fact, our results in Fig 4(b) show
> that the technique works very well on the convolutional outputs.
> We believe the difference is due to the normalization of the LIME weights.
> The normalization we introduce effectively looks at a series of candidate im-
> portance distributions (local models), and averages them after applying locality
> factors. This turns out to be equivalent roughly to looking at squared impor-
> tance factors modulo some normalization.  We believe this tends to discount
> importance of features more steeply, which makes it more difficult for a feature
> to be considered “important”. I think we observe a strong effect in the human
> evaluation because we look at class interpretations, and other methods are ori-
> ented around generating a single local explanation, so they are not as equipped
> for dealing with aggregating multiple interpretations.
> Thank you for your additional comments, however I am not quite sure I
> understand. From my understanding it is not quite clear how to apply KAR
> to test the source of the difference in methods. KAR simply proposes that you
> measure the performance loss in a model induced by discarding certain features.
> We applied it to the quantitative evaluation to show that the features retained
> by NormLIME tended to lead to better performance (accuracy) over those from
> other methods. We don’t apply KAR in the human evaluation because it is
> inherently a qualitative evaluation of human perception.
> Regarding the number of samples of LIME / NormLIME vs SG and VG,
> this is a mistaken omission. For all explainers in the human user experiments,
> we generated each class explanation from aggregating n=10 individual interpre-
> tations respectively. In terms of the specific parameters for LIME we use the
> default number of samples suggested by Ribeiro et. al., and for NormLIME each
> individual local LIME explanation again uses the suggested default parameters

---

### Decision · Program_Chairs · 2019-12-19

**Decision:**

Reject

**Comment:**

The paper aims to extract the set of features explaining a class, from a trained DNN classifier.

The proposed approach relies on LIME (Ribeiro et al. 2016), modified as follows: i) around a point x, a linearized sparse approximation of the classifier is found (as in LIME); ii) for a given class, the importance of a feature aggregates the relative absolute weight of this feature in the linearized sparse approximations above; iii) the explanation is made of the top features in terms of importance.

This simple modification yields visual explanations that significantly better match the human perception than the SOTA competitors.

The experimental setting based on the human evaluation via a Mechanical Turk setting is the second contribution of the approach. The feature importance measure is also assessed along a Keep and Retrain mechanism, showing that the approach selects actually relevant features in terms of prediction.
Incidentally, it would be good to see the sensitivity of the method to parameter $k$ (in Eq. 1).

As noted by Rev#1, NormLIME is simple (and simplicity is a strength) and it demonstrates its effectiveness on the MNIST data. However, as noted by Rev#4, it is hard to assess the significance of the approach from this only dataset.

It is understood that the Mechanical Turk-based assessment can only be used with a sufficiently simple problem.  However, complementary experiments on ImageNet for instance, e.g., showing which pixels are retained to classify an image as a husky dog, would be much appreciated to confirm the merits and investigate the limitations of the approach.